

# Importance of considering interoceptive abilities in alexithymia assessment

Alicia Fournier[1,2], Olivier Luminet[3,4], Michael Dambrun[5], Frédéric Dutheil[5,6,7], Sonia Pellissier[8] and Laurie Mondillon[5]

[1] Laboratory Psy-DREPI, Université de Bourgogne, Dijon, France
[2] Behaviors, Risk and Health, CNRS, MSHE Claude-Nicolas Ledoux, Besançon, France
[3] Research Institute for Psychological Sciences, Université Catholique de Louvain, Louvain-La-Neuve, Belgium
[4] Belgium Fund for Scientific Research (FRS-FNRS), Brussels, Belgium
[5] Laboratoire de Psychologie Sociale et Cognitive (LAPSCO), CNRS UMR 6024, Université Clermont Auvergne, Clermont-Ferrand, France
[6] Preventive and Occupational Medicine, University Hospital of Clermont-Ferrand, Clermont-Ferrand, France
[7] Australian Catholic University, Melbourne, VIC, Australia
[8] Laboratoire Inter-Universitaire de Psychologie, Personnalité, Cognition et Changement Social (LIP/PC2S), Université Savoie Mont Blanc, Chambéry, France

Corresponding authors
Alicia Fournier,
alicia.fournier@u-bourgogne.fr
Laurie Mondillon,
laurie.mondillon@uca.fr

## ABSTRACT

**Background:** Recent studies have shown that people with high alexithymia scores have decreased interoceptive abilities, which can be associated with psychological and physical disorders. Early assessments of the alexithymia trait included the evaluation of these abilities through the dimension measuring the difficulty in identifying and distinguishing between feelings and bodily sensations (the 26-item Toronto Alexithymia Scale; TAS-26). The revised version of the TAS, the TAS-20, contains a three-factor solution that does not involve a dimension assessing interoceptive abilities. However, the three items allowing the evaluation of these abilities are still present in the TAS-20. In this context, we hypothesized that the 3 items which assess interoceptive abilities in the TAS-20 should constitute an independent factor. In addition to exploring the internal structure of the TAS-20, we examined its external validity by assessing the relationships between the new factors and self-reported measures of personality trait and psychological and physical health.

**Method:** Two online studies ($N = 253$ and $N = 287$) were performed. The participants completed the TAS-20 and a set of psychological questionnaires (e.g., anxiety, depression) and health questions (e.g., "Do you suffer from a somatic disorder?"). The structure of the TAS-20 was examined using exploratory factor analysis (EFA), followed by an investigation of the relationships between the resulting new factors and other psychological and health data using regressions. In both studies, EFA revealed a new structure of the questionnaire consisting of four dimensions: (1) difficulty in the awareness of feelings, (2) externally oriented thinking, (3) difficulty in interoceptive abilities, and (4) poor affective sharing. The first factor was positively associated with all self-reported psychological and personality trait measures while the third factor was associated more with somatic disorders and medication intake.

**Results:** Our results suggest the presence of a new latent factor in the assessment of alexithymia that reflects interoceptive abilities specifically related to health and

personality trait outcomes. In accordance with the results and the literature, it seems important to include an assessment of interoceptive abilities when considering the evaluation of alexithymia. The next step would be to develop a valid measure of these abilities.

## INTRODUCTION

The alexithymia construct, literally meaning "without words for feelings" (*Apfel & Sifneos, 1979*), is derived from clinical observations of patients suffering from psychosomatic disorders (*MacLean, 1949*; *Marty & De M'Uzan, 1963*; *Ruesch, 1948*). Based on these observations, three main features were defined for the alexithymia construct: (i) difficulty identifying and describing one's own feelings, (ii) limited imaginative processes, and (iii) an externally oriented cognitive style (*Apfel & Sifneos, 1979*; *Nemiah & Sifneos, 1970*). Following a review of the literature on alexithymia, the features "*difficulty in distinguishing between feelings and bodily sensations*" and "*social conformity*" were added later (*Taylor, Ryan & Bagby, 1985*).

Alexithymia is associated with many psychological and physical disorders, such as anxiety (*Karukivi et al., 2015*), depression (*Li et al., 2015*), somatization (*Brandt, Pintzinger & Tran, 2015*), somatic complaints (*Tominaga et al., 2013*), eating disorders (*Jenkinson, Taylor & Laws, 2018*), myocardial infarction (*Silva et al., 2016*), carotid atherosclerosis (*Grabe et al., 2010*), and higher mortality rates (*Tolmunen et al., 2010*). For this reason, alexithymia is a construct of interest in many theoretical models of health psychology (*Lumley, Neely & Burger, 2007*). It is, therefore, necessary to correctly assess this construct using reliable and valid measures. Different scales have been developed to evaluate the alexithymia construct, such as the Beth Israel Hospital Questionnaire (*Sifneos, 1973*), the Schalling–Sifneos Personality Scale (*Apfel & Sifneos, 1979*), and the MMPI alexithymia scale (*Kleiger & Kinsman, 1980*), but they have inadequate psychometric qualities (*Taylor, Ryan & Bagby, 1985*). Consequently, the 26-item Toronto Alexithymia Scale (TAS-26) was developed. This scale assessed four dimensions: (i) difficulty identifying and distinguishing between feelings and bodily sensations (DIDF), (ii) difficulty describing feelings (DDF), (iii) reduced daydreaming, and (iv) externally oriented thinking (EOT) (*Taylor, Ryan & Bagby, 1985*). Due to problems with the compositional structure of the TAS, the revision of this scale led to the development of the 23-item TAS (TAS-R), which assessed two dimensions: (i) ability to distinguish between feelings and bodily sensations associated with emotional arousal and the ability to describe feelings to others, and (ii) EOT (*Taylor, Bagby & Parker, 1992*). During this review, the *reduced daydreaming* dimension was suppressed due to low corrected item-total correlations with the full TAS and a negative correlation with the DIDF factor. In addition, the DIDF and DDF dimensions were merged into one dimension, and one item was replaced by a new one. Two EOT items were removed and five new items were added.

Subsequently, due to social desirability response bias and a lack of inter-correlations between factors, the TAS was reviewed one more time, resulting in the 20-item TAS (TAS-20) (*Bagby, Parker & Taylor, 1994*; *Taylor, Bagby & Parker, 2003*). The DDF dimension was reintroduced as a distinct dimension. The DIDF dimension became a difficulty identifying feelings (DIF) dimension and two items were deleted. However, the two items which were removed did not refer to the body, and two other items related to interoception were retained. The notion of *difficulty distinguishing between feelings and bodily sensations* was therefore dropped in the DIF dimension. In addition, one item from the EOT was suppressed.

The TAS-20 is the most widely used scale for measuring alexithymia both in empirical research and for clinical assessment (*Lane et al., 2015*; *Sekely, Bagby & Porcelli, 2018*). Despite the fact that the alexithymia concept covers more features, this scale assesses three dimensions: (i) DIF, (ii) DDF, and (iii) EOT. The TAS-20 has good reliability and factorial validity in different languages and cultures (*Taylor, Bagby & Parker, 2003*), and the three-dimensional model is considered as the best fit (*Bagby, Parker & Taylor, 1994*; *Loas et al., 1997*; *Parker et al., 1993*). However, several studies have reported, for various reasons, that the factor structure of this scale is not always consistent (*Haviland & Reise, 1996*; *Kooiman, Spinhoven & Trijsburg, 2002*; *Müller, Bühner & Ellgring, 2003*). First, the EOT dimension might better reflect the social norms that guide emotional behaviors rather than a cognitive style of thinking (*Dere et al., 2013*). This probably leads to the lack of internal consistency of this dimension (*Bressi et al., 1996*; *Taylor, Bagby & Parker, 2003*; *Zhu et al., 2007*). Second, the verbalization and the differentiation of feelings seem theoretically interconnected (*Lane & Schwartz, 1987*), which explains why some studies found a unique factor that combines the DIF and DDF dimensions (*Erni, Lötscher & Modestin, 1997*; *Kooiman, Spinhoven & Trijsburg, 2002*; *Loas et al., 1996*). Third, the lack of consistency could be due to the analysis performed (*Loas et al., 2001*). Indeed, most studies that have reported other solutions than the three-factor solution only used exploratory factorial analysis (EFA), which accounts more for the existence of alternative models (*Loas et al., 2001*), whereas the appropriate tool to confirm the three-factor solution seems to be confirmatory factorial analysis (CFA). Fourth, we assume that this could be due to the suppression of the *difficulty distinguishing between feelings and bodily sensations* label without the suppression of items referring to bodily sensations. This rearrangement could lead to the existence of a latent factor in the TAS-20, which could reflect the old structure of the TAS-26 and TAS-R.

## Overview

Therefore, the aim of this paper was to examine the structure of the TAS-20 using EFA in a subclinical population. Contrary to the opinion of *Loas et al. (2001)*, we decided to use EFA and not CFA. In fact, the aim of this paper was not to validate or confirm the factor structure of this scale but to explore whether the TAS-20 contains a latent factor assessing interoceptive abilities. The use of CFA would have involved making choices based on theoretical data, which could influence the results of our exploratory studies. Two different versions of the TAS-20 are available in French. The first comprises items rated on a 4-point

Likert scale (*Bruchon-Schweitzer, 2002*) and the second contains items rated on a 5-point Likert scale (*Loas et al., 1996*). There are no other differences between the two versions; one is the gold standard (5-point Likert scale) and the other is available in French, but is not widely used (4-point Likert scale). In the first study, the participants completed the French version of the TAS-20 with a 4-point Likert scale and in the second study, the participants completed the French version of the TAS-20 with a 5-point Likert scale. In addition to exploring the internal structure of the TAS-20, we examined its external validity by assessing the relationships between the new alexithymia factors resulting from the EFA and personality trait and the indicators of psychological and physical health. Alexithymia "is a marker of atypical interoception" (*Murphy et al., 2017*) and the link between alexithymia and interoception is currently receiving significant attention in the field, as shown by several articles published in recent years (*Bornemann & Singer, 2017*; *Brewer, Cook & Bird, 2016*; *Murphy, Catmur & Bird, 2018*; *Zamariola et al., 2018*, *2019*). Difficulties in interoceptive abilities can be associated with psychological and physical impairments (*Murphy et al., 2017*), and moderately with an emotional instability, a feature of the neuroticism trait (*Fiene, Ireland & Brownlow, 2018*; *Kanbara & Fukunaga, 2016*). For these reasons, we assumed that if a latent factor existed in the TAS-20 that assesses interoceptive abilities, it should be associated, like other alexithymia dimensions, with the presence of physical and psychological health problems and the personality trait (i.e., emotional instability). If alexithymia is also defined by difficulties in interoceptive abilities, and these difficulties are related to health disorders and personality trait, this will allow new ways of theoretical reflection to be explored in order to understand the mechanisms underlying the development of DIF and DDF and to consider new interventions to reduce the health impact of alexithymia. In particular, for some individuals, it may be that DIF and DDF are mediated by difficulties in interoceptive abilities, which are abilities essential to emotion knowledge (*Luminet & Zamariola, 2018*). If individuals cannot correctly perceive and interpret their bodily sensations, they may have difficulty identifying and describing their feelings, but also regulating them when necessary. In the long term, this could have several impacts on an individual's well-being. We would like to point out that the purpose of this work is not to criticize the factorial structure of the TAS-20, but rather to highlight the possible existence of a latent factor that could assess difficulties in interoceptive abilities, as stated in the earlier versions of the questionnaire (TAS-26 and TAS-R). The first aim of this paper was to perform exploratory factor analysis (EFA) to investigate the possible existence of a latent factor that could assess difficulties in interoceptive abilities, as stated in the earlier versions of the questionnaire (TAS-26 and TAS-R). We carried out exploratory studies on the factorial structure of the TAS-20. We assumed that the TAS-20 contained a latent factor to assess individuals' interoceptive abilities during emotional arousal. The presence of this factor could reflect the initial structure of the TAS proposed by *Taylor, Ryan & Bagby (1985)*, which contained the DIDF factor. Additionally, based on the literature on interoception, we assumed that if there is a latent factor for assessing interoceptive abilities, it would be related to health and personality outcomes. In particular, it would be associated with high scores of depression, anxiety, perceived stress and emotional instability.

## MATERIALS AND METHODS

### Participants and ethics statement

Overall, 540 participants (Study 1: $N$ = 253; Study 2: $N$ = 287) were enrolled. We recruited 395 undergraduate psychology students (Study 1: 16 men, 92 women; mean age: 19.44 ± 1.28; Study 2: 35 men, 252 women; mean age: 19.56 ± 1.58) from Clermont Auvergne University (formerly Blaise Pascal University, Clermont-Ferrand, France). The other participants were recruited on a voluntary basis from the general population through social networks (61 men, 84 women; mean age: 37.26 ± 14.03). Table 1 provides an overview of the descriptive statistics of the samples and the measures used in each study. The Ethics Committee in Clermont-Ferrand approved the study protocol (CPP SUD-EST 6, IRB00008526, 2015-CE23). The nature and potential risks of the study were fully explained to the participants. Written informed consent was obtained from each participant. The experimental data is available at https://osf.io/8kncz.

### Measures

*Alexithymia* was assessed using the 20-item Toronto Alexithymia Scale (TAS-20) (*Bagby, Parker & Taylor, 1994*). The 20 items of this scale evaluate three dimensions of alexithymia: (a) DIF (items 1, 3, 6, 7, 9, 13, 14) (Study 1: α = 0.81; Study 2: α = 0.84), (b) DDF (items 2, 4, 11, 12, 17) (Study 1: α = 0.80; Study 2: α = 0.79), and (c) EOT (items 5, 8, 10, 15, 16, 18, 19, 20) (Study 1: α = 0.65; Study 2: α = 0.59). Items 4, 5, 10, 18, and 19 are reverse coded. Two versions of the TAS-20 exist in French. In Study 1, we used the French version with items rated on a 4-point scale ranging from 1 (*rarely*) to 4 (*very often*) (*Bruchon-Schweitzer, 2002*). The total score varies from 20 to 80, with a high score indicating a high level of alexithymia (α = 0.83). In Study 2, we used the French version with items measured on a 5-point Likert scale (*Loas et al., 1996*) ranging from 1 (*strongly disagree*) to 5 (*strongly agree*). The total score varies from 20 to 100, with a high score indicating a high level of alexithymia (α = 0.84). With the 4-point scale, a score >49 indicates an alexithymic trait, and with the 5-point scale, a score >61 indicates an alexithymic trait.

    *Trait anxiety* was assessed using the Trait subscale of the State-Trait Anxiety Inventory (STAI-T) (*Bruchon-Schweitzer & Paulhan, 1993*; *Spielberger et al., 1983*) consisting of 20 items measured on a scale ranging from 1 (*almost never*) to 4 (*almost always*). Items 1, 3, 6, 7, 10, 13, 14, 16, and 19 are reverse coded. The overall score varies from 20 to 80, with higher scores indicating a high level of anxiety (Study 1: α = 0.91; Study 2: α = 0.91). A score >65 is considered very high.

    In Study 1, *depressive symptomatology* was assessed using the Depression subscale of the Hospital Anxiety and Depression Scale (HADS-D) (*Lepine, Godchau & Brun, 1985*; *Zigmond & Snaith, 1983*). The seven items of this scale use a 4-point scale (from 0 to 3) to measure the symptoms or behaviors that are often associated with depression. This scale was validated in a variety of populations, including the general population, general practice and psychiatric patients (*Bjelland et al., 2002*). The total score varies from 0 to 21, with a high score indicating a high level of depressive symptomatology. A score >8

**Table 1 Socio-demographic, general health, and psychological data for both samples.**

| | Total sample Study 1 | Total sample Study 2 | p-value[a] |
|---|---|---|---|
| Socio-demographic data | | | |
| Number of participants | 253 | 287 | |
| Percentage of Women | 69.57% | 87.8% | <0.001*** |
| Age | 29.65 ± 13.82 | 19.56 ± 1.58 | <0.001*** |
| Health data | | | |
| Cardiovascular disease, N (% of sample) | 13 (5.14) | 13 (4.53) | 0.841 |
| Eating disorders, N (% of sample) | 24 (9.49) | 22 (7.66) | 0.537 |
| Somatic disorders, N (% of sample) | 11 (4.35) | 12 (4.18) | 1 |
| Medication intake, N (% of sample) | 34 (13.44) | 32 (11.14) | 0.43 |
| Anxiolytics | 3 | 5 | |
| Antidepressants | 6 | 4 | |
| Anti-inflammatory drugs | 2 | 1 | |
| Antihistamines | 2 | 5 | |
| Migraine medications | 1 | 4 | |
| Asthma medications | 4 | 1 | |
| Others[1] | 19 | 16 | |
| Psychological data | | | |
| TAS-20 (/100) | 48.58 ± 10.87 | 51.59 ± 11.62 | 0.002** |
| DIF (/35) | 15.09 ± 5.02 | 18.79 ± 6.33 | <0.001*** |
| DDF (/25) | 13.41 ± 4.53 | 15.19 ± 4.70 | <0.001*** |
| EOT (/40) | 20.08 ± 4.70 | 17.61 ± 4.15 | <0.001*** |
| STAI-T (/80) | 42.14 ± 9.66 | 45.53 ± 9.71 | <0.001*** |
| HADS-D (/21) | 3.71 ± 2.90 | – | |
| BDI-13 (/39) | – | 18.85 ± 4.56 | |
| BFI-N (/40) | 21.02 ± 6.94 | 23.62 ± 6.80 | <0.001*** |
| PSS (/40) | 26.74 ± 7.48 | 29.40 ± 7.47 | <0.001*** |
| Brief Cope | | | |
| Functional coping (/8) | 5.20 ± 1.30 | 5.14 ± 1.10 | 0.535 |
| Coping with varying functionality (/8) | 4.43 ± 1.02 | 4.52 ± 0.92 | 0.279 |
| Dysfunctional coping (/8) | 3.03 ± 0.86 | 3.22 ± 0.83 | 0.008** |

Notes:
** $p < 0.01$.
*** $p < 0.001$.
Data represents means ± SD.
[a] Regards differences between Study 1 and Study 2; ANOVA Test or Chi2–Test. To compare the samples, alexithymia scores from Study 1 were transformed into a 5-point Likert scale.
[1] The "others" category included drugs with a low frequency of use such as beta-blocker or immunosuppressant. The detailed list of medications may include multiple intakes. The same participant could be included in two categories of drugs.

identifies those with a positive history of depression. The HADS-D evaluates moderate depressive states; thus, it does not mention suicidal ideation (*Hansson et al., 2009*) (α = 0.73). However, in Study 2, we assessed the depression score using the 13-item Beck Depression Inventory (BDI-13) (*Beck et al., 1961*; *Collet & Cottraux, 1986*). This scale measure the depression symptomatology on a 4-point scale from 0 to 3. The total score

varies from 0 to 39, with a high score indicating a high level of depressive symptomatology ($\alpha = 0.79$). A score ≥30 indicates severe depression. Because the BDI-13 contains an item on suicidal ideologies, we were not permitted, for ethical reasons, to use this scale in Study 1, which included individuals from the general population.

*Emotional instability* was measured using the Neuroticism dimension of the Big Five Inventory (BFI-N) (*John, Donahue & Kentle, 1991*; *Plaisant et al., 2010*). The neuroticism trait is defined by negative affectivity, such as emotional instability, anger, worry, and sadness (*Weiss & Costa, 2005*). Moreover, this trait is positively associated with alexithymia and somatization (*Porcelli & Taylor, 2018*). The neuroticism dimension comprises 8 items rated on a 5-point scale ranging from 1 (*not agree at all*) to 5 (*completely agree*). Items 9, 24, and 34 are reverse coded. The total score varies from 5 to 40, with a high total score indicating a high emotional instability (Study 1: $\alpha = 0.85$; Study 2: $\alpha = 0.84$). As the BFI is not a diagnostic instrument, there is no cutoff.

*Perceived stress* was evaluated using the Perceived Stress Scale (PSS) (*Bellinghausen et al., 2009*; *Cohen & Williamson, 1988*). The 10 items of this scale measure the degree to which everyday life situations are appraised as stressful on a 5-point scale ranging from 0 (*never*) to 4 (*very often*). Items 4, 5, 7, and 8 are reverse coded. The total score varies from 0 to 40, with a high score indicating a high level of perceived stress (Study 1: $\alpha = 0.88$; Study 2: $\alpha = 0.88$). As the PSS is not a diagnostic instrument, there is no cutoff.

*Coping strategies* were assessed using the Brief Cope (*Carver, 1997*; *Muller & Spitz, 2003*). The 28 items of this scale measure fourteen coping strategies on a 4-point scale ranging from 1 (*not at all*) to 4 (*absolutely*). Due to a large number of coping styles, and in accordance with *Muller & Spitz (2003)*, we grouped them into three categories: (a) functional coping (mean of active coping, planning, positive reframing, and acceptance) (Study 1: $\alpha = 0.86$; Study 2: $\alpha = 0.80$), (b) dysfunctional coping (mean of denial, behavioral disengagement, substance use, and self-blame) (Study 1: $\alpha = 0.73$; Study 2: $\alpha = 0.69$), and (c) coping with varying functionality (mean of self-distraction, humor, venting, use of emotional support, use of instrumental support, and religion) (Study 1: $\alpha = 0.81$; Study 2: $\alpha = 0.72$). Coping with varying functionality are strategies that are less likely to be delimited as functional or dysfunctional because they depend on the circumstances. For example, use of emotional support can either help or harm the resolution of a stressful situation (*Montgomery, Demers & Morin, 2010*). For these reasons, these coping strategies were not used in subsequent analyses. We then created a difference score between functional and dysfunctional coping (F–D) to highlight the use of appropriate and effective strategies. This score suggests that the larger and more positive the difference between the scores, the more individuals use functional coping strategies, and the larger and more negative the difference between scores, the more individuals use dysfunctional coping.

## Other health assessments

Participants were asked if they were currently under medical treatment. Individuals who reported medication intake had to specify the type of medication. In addition, when the participants completed the survey they were asked if they currently had cardiovascular

disease or hypertension, chronic visceral disease or a somatic disorder (e.g., eczema, asthma, headaches, somatoform disorders, colitis), or an eating disorder. Medication intake, cardiovascular diseases, somatic disorders, and eating disorders were measured on a binary scale coded as 1 (yes) and 0 (no). These data were treated in a qualitative way.

## Procedure

After providing informed consent, the participants completed a set of online questionnaires via the LimeSurvey platform. The students completed the protocol at the university while individuals from the general population completed the surveys at home. All participants completed the questionnaires in the following order: (1) STAI-T, (2) TAS-20, (3) HADS-D (Study 1) /BDI-13 (Study 2), (4) BFI-N, (5) PSS, and (6) Brief cope. Finally, the participants had to complete the four binary questions about health measures and demographic data.

## Statistical analysis

First, we performed descriptive statistics of the health and psychological data. The aim of this paper was not to compare the samples with each other but to verify whether our results were consistent across studies. However, for information purposes, comparative analyses were performed using ANOVAs to check for differences between samples in socio-demographic, psychological, and health measures. To compare alexithymia scores between samples, we transformed the 4-point Likert scale used in Study 1 to a 5-point Likert scale for each item and calculated the dimensions of alexithymia (i.e., DIF, DDF, EOT).
For the depression score, we used two different scales (depression subscale of the HADS in Study 1 and BDI-13 in Study 2) that assess different features of depression; therefore, we could not compare depression scores between samples. When the homoscedasticity assumption was violated, we used adjusted *Welch's F*. We performed Pearson chi-square tests to compare gender and health data between studies.

We performed an EFA with direct Oblimin rotation and principal axis factoring to examine the factorial structure of the TAS-20. To verify the sampling adequacy for the analysis, we computed the Kaiser-Meyer-Olkin (KMO) values for all individual items. For each factor, we estimated the internal consistency using Cronbach's alpha. Based on those results, in both Study 1 and Study 2, we decided to omit items 16 and 20, which belonged to a factor with very low reliability. These two items represent the preference for entertainment rather than an exploration of a deeper meaning in movies or plays. We then conducted another EFA to examine the structure of the remaining 18 items, and we called the resulting factors the "*latent factors*" (LF). Any items with component loadings <0.30 were considered as explaining only a small part of the factor (*Field, 2013*). We decided not to report the value of these component loadings in the Tables unless items explained the factor. Reliability analysis was carried out for each component. We considered Cronbach's alphas <0.50 as not satisfactory (*Taber, 2018*).

Finally, we performed multivariate regression analyses to examine whether the LFs were predictive of the psychological and health measures. Specifically, we conducted multivariate logistic regressions on each (binomial) health-related measure (somatic

disorders, eating disorders, cardiovascular diseases, and medication intake), and multivariate linear regressions on each psychological measure (anxiety, depression, emotional instability, perceived stress, and coping strategies).

Statistical analysis was carried out using SPSS version 24.0 for Macintosh (Statistical Package for the Social Sciences; IBM Corporation, Armonk, NY, USA). The $p$-value for statistical significance was set at $p < 0.05$, and the trend for significance was set at $p < 0.07$.

## RESULTS

Descriptive statistics are presented in Table 1. When comparing samples from Study 1 and Study 2, we found that the participants in Study 2 included more women ($\chi^2$ *Pearson* (1) = 27.21, $p < 0.001$) and younger (*Welch F*(1,257.83) = 133.29, $p < 0.001$), more anxious ($F$(1,538) = 16.52, $p < 0.001$), emotionally more unstable ($F$(1,538) = 19.28, $p < 0.001$) individuals who perceived more stress ($F$(1,538) = 17.06, $p < 0.001$) and used more dysfunctional coping strategies ($F$(1,538) = 6.99, $p = 0.008$) compared to participants in Study 1. Moreover, the participants in Study 2 scored higher on alexithymia (TAS-20) compared to participants in Study 1 ($F$(1,538) = 9.58; $p = 0.002$) and had more difficulty identifying (*Welch F*(1,532.24) = 57.35; $p < 0.001$) and describing feelings ($F$(1,538) = 19.92; $p < 0.001$). However, the participants in Study 2 had less EOT compared to the participants in Study 1 ($F$(1,538) = 42.09; $p < 0.001$).

For more information on the recruited population, we also reported the number of individuals with psychological disorders. In Study 1, 34 participants were considered alexithymic (>49) ($M = 54.06 \pm 4.42$), 18 participants had a positive history of depression (>8) ($M = 10.89 \pm 2.99$), and 3 participants had a very high anxiety score (>65) ($M = 71.67 \pm 5.51$). In Study 2, 64 participants were considered alexithymic (>61) ($M = 67.22 \pm 4.44$), 10 participants had a severe depression (≥30) ($M = 30.9 \pm 1.29$), and 6 participants had a very high anxiety score (>65) ($M = 67.17 \pm 1.94$).

### Exploratory factor analysis of the TAS-20

In Study 1, the KMO verified the sampling adequacy for the analysis (*KMO* = 0.85; individual *KMO* values ≥0.55 and ≤0.94). The Kaiser criterion indicated five factors (F1–F5), which accounted for 57.54% of the total variance. F1 consisted of seven items (items 1, 2, 6, 9, 11, 13, 14; $\alpha_1 = 0.85$), F2 consisted of five items (items 5, 8, 10, 18, 19; $\alpha_2 = 0.62$), F3 consisted of two items (items 3, 7; $\alpha_3 = 0.66$), F4 consisted of two items (items 16, 20; $\alpha_4 = 0.36$), and F5 consisted of four items (items 4, 12, 15, 17; $\alpha_5 = 0.68$) (Table 2).

F4, which included item 16 (*A preference for entertainment shows rather than psychological dramas*) and item 20 (*A preference not to search for the hidden meanings of films or plays in order to not distract from the pleasure*), had very low reliability ($\alpha_4 = 0.36$). Therefore, we conducted another EFA without these items. The KMO verified the sampling adequacy (*KMO* = 0.87; individual *KMO* values ≥0.66 and ≤0.90). The results revealed a new structure with four latent factors, LF1 (items 1, 2, 6, 9, 11, 13, 14; $\alpha_1 = 0.85$), LF2 (items 5, 8, 10, 18, 19; $\alpha_2 = 0.62$), LF3 (items 3, 7; $\alpha_3 = 0.66$), and LF4 (items 4, 12, 15, 17; $\alpha_4 = 0.68$), which accounted for 56% of the total variance. This reallocation was

**Table 2 Loadings after Oblimin rotation from the EFA of the TAS-20, from the EFA of the TAS without items 16 and 20, and comparative attribution of items in Study 1.**

| Items | Factor (F) | | | | | Latent factor (LF) | | | | Theoretical attribution | New attribution |
|---|---|---|---|---|---|---|---|---|---|---|---|
| | F1 | F2 | F3 | F4 | F5 | LF1 | LF2 | LF3 | LF4 | | |
| 1 | **0.75** | | | | | **0.76** | | | | DIF | LF1 |
| 2 | **0.58** | | | | 0.41 | **0.60** | | | 0.37 | DDF | LF1 |
| 3 | | | **0.57** | | | | | **0.57** | | DIF | LF3 |
| 4 | | | | **0.45** | | 0.38 | | | **0.44** | DDF | LF4 |
| 5 | | **0.29** | | | | | **0.38** | | | EOT | LF2 |
| 6 | **0.59** | | | | | **0.58** | | | | DIF | LF1 |
| 7 | | | **0.74** | | | | | **0.77** | | DIF | LF3 |
| 8 | | **0.24** | | | | | **0.33** | | | EOT | LF2 |
| 9 | **0.75** | | | | | **0.74** | | | | DIF | LF1 |
| 10 | | **0.61** | | | | | **0.59** | | | EOT | LF2 |
| 11 | **0.51** | | | | | **0.53** | | | | DDF | LF1 |
| 12 | | | | | **0.50** | 0.30 | | | **0.47** | DDF | LF4 |
| 13 | **0.69** | | | | | **0.72** | | | | DIF | LF1 |
| 14 | **0.47** | | | | | **0.45** | | | | DIF | LF1 |
| 15 | | 0.35 | | | **0.41** | | 0.37 | | **0.41** | EOT | LF4 |
| 16 | | | **0.41** | | | – | – | – | – | EOT | – |
| 17 | | | | | **0.61** | | | | **0.65** | DDF | LF4 |
| 18 | | **0.35** | | | | | **0.37** | | | EOT | LF2 |
| 19 | | **0.71** | | | | | **0.73** | | | EOT | LF2 |
| 20 | | | | **0.68** | | – | – | – | – | EOT | – |
| Eigenvalues | 5.49 | 2.15 | 1.55 | 1.20 | 1.12 | 5.47 | 2.06 | 1.47 | 1.09 | | |
| % of variance | 27.44 | 10.76 | 7.74 | 5.98 | 5.61 | 30.37 | 11.43 | 8.17 | 6.04 | | |
| α | 0.85 | 0.62 | 0.66 | 0.36 | 0.68 | 0.85 | 0.62 | 0.66 | 0.68 | | |

Note:
Factor loadings are highlighted in bold type. For easy reading, all values of loading <0.30 were not reported, except if they explained the factor.

conceptually relevant, since LF1 (5 items of DIF and 2 items of DDF) referred to difficulty in the awareness of feelings, LF2 (5 items of EOT) referred to EOT, LF3 (2 items of DIF) referred to difficulty in interoceptive capacities, and LF4 (3 items of DDF and 1 item of EOT) referred to poor affective sharing (Table 2).

In Study 2, the KMO verified the sampling adequacy for the analysis ($KMO = 0.86$; individual $KMO$ values $\geq 0.59$ and $\leq 0.92$). The Kaiser criterion indicated six factors (F1–F6), which accounted for 62.57% of the total variance. F1 consisted of seven items (items 1, 2, 4, 6, 9, 13, 14; $\alpha_1 = 0.88$), F2 consisted of three items (items 10, 18, 19; $\alpha_2 = 0.51$), F3 consisted of four items (items 11, 12, 15, 17; $\alpha_3 = 0.70$), F4 consisted of three items (items 3, 7, 13[2]; $\alpha_4 = 0.71$), F5 consisted of two items (items 16, 20; $\alpha_5 = 0.43$), and F6 consisted of two items (items 5, 8; $\alpha_6 = 0.37$) (Table 3).

F5, which included item 16 (*A preference for entertainment shows rather than psychological dramas*) and item 20 (*A preference not to search for the hidden meanings of films or plays in order not to distract from the pleasure*), had a low reliability ($\alpha_5 = 0.43$) and

**Table 3 Loadings after Oblimin rotation from the EFA of the TAS without items 16 and 20, and comparative attribution of items in Study 2.**

| Items | Factor (F) | | | | | | Factor (F) | | | | | Latent factor (LF) | | | | Theoretical attribution | New attribution |
|---|---|---|---|---|---|---|---|---|---|---|---|---|---|---|---|---|---|
| | F1 | F2 | F3 | F4 | F5 | F6 | F1 | F2 | F3 | F4 | F5 | LF1 | LF2 | LF3 | LF4 | | |
| 1 | **0.72** | | | | | | **0.72** | | | | | **0.81** | | | | DIF | LF1 |
| 2 | **0.69** | | | | | | **0.81** | | | | | **0.81** | | | | DDF | LF1 |
| 3 | | | **0.68** | | | | | | **0.64** | | | | | **0.62** | | DIF | LF3 |
| 4 | **0.59** | | | 0.31 | | | **0.71** | | | | | **0.69** | | | | DDF | LF1 |
| 5 | | | | | | −0.67 | | 0.46 | | | | | **0.43** | | | EOT | LF2 |
| 6 | **0.62** | | | | | | **0.45** | | | | 0.36 | **0.54** | | | | DIF | LF1 |
| 7 | | | **0.65** | | | | | | **0.72** | | | | | **0.65** | | DIF | LF3 |
| 8 | | | | | | −0.32 | | 0.20 | | | | | 0.20 | | | EOT | LF2 |
| 9 | **0.65** | | | | | | **0.73** | | | | | **0.76** | | | | DIF | LF1 |
| 10 | | **0.50** | | | | | | 0.54 | | | | | **0.53** | | | EOT | LF2 |
| 11 | | | −0.59 | | | | 0.33 | | −0.56 | | | | | | −0.59 | DDF | LF4 |
| 12 | | | −0.57 | | | | | | −0.57 | | | | | | −0.60 | DDF | LF4 |
| 13 | **0.37** | | | 0.37 | | | 0.33 | | | 0.40 | | **0.38** | | 0.35 | | DIF | LF1 |
| 14 | **0.61** | | | | | | **0.45** | | | | 0.40 | **0.56** | | | | DIF | LF1 |
| 15 | | | −0.44 | | | | | | −0.49 | | | | | | −0.45 | EOT | LF4 |
| 16 | | | | −0.41 | | | – | – | – | – | – | – | – | – | – | EOT | – |
| 17 | | | −0.70 | | | | | | −0.62 | | | | | | −0.63 | DDF | LF4 |
| 18 | | **0.40** | | | | | | 0.43 | | | | | **0.44** | | | EOT | LF2 |
| 19 | | **0.62** | | | | | | 0.54 | | | | | **0.55** | | | EOT | LF2 |
| 20 | | | | −0.40 | | | – | – | – | – | – | – | – | – | – | EOT | – |
| Eigenvalues | 5.57 | 2.07 | 1.49 | 1.20 | 1.12 | 1.07 | 5.51 | 1.87 | 1.47 | 1.10 | 1.02 | 5.51 | 1.87 | 1.10 | 1.47 | | |
| % of variance | 27.85 | 10.36 | 7.43 | 6.00 | 5.59 | 5.34 | 30.61 | 10.39 | 8.14 | 6.13 | 5.67 | 30.61 | 10.39 | 6.13 | 8.14 | | |
| α | 0.88 | 0.51 | 0.70 | 0.71 | 0.43 | 0.37 | 0.86 | 0.51 | 0.70 | 0.71 | – | 0.88 | 0.51 | 0.64 | 0.70 | | |

**Note:**
Factor loadings are highlighted in bold type. For easy reading, all values of loading <0.30 were not reported, except if they explained the factor.

seemed to have limited relevance to the alexithymia trait. Hence, we decided to omit them. Moreover, F6, which included item 5 (*A preference to analyze problems rather than describe them*) and item 8 (*A preference to let things happen rather than to understand*) also had very low reliability ($\alpha_6 = 0.37$). However, these items seemed to be representative of the alexithymia trait. Because alexithymia reflects a tendency to focus on the concrete details of external events rather than on feelings, suppressing them would have been against the theory. Therefore, we decided to keep them.

As in Study 1, we conducted another EFA without items 16 and 20. The KMO verified the sampling adequacy (*KMO* = 0.87; individual *KMO* values ≥0.59 and ≤0.92). The results showed a new structure with five factors, F1 (items 1, 2, 4, 6, 9, 14; $\alpha_1 = 0.86$), F2 (items 5, 8, 10, 18, 19; $\alpha_2 = 0.51$), F3 (items 11, 12, 15, 17; $\alpha_3 = 0.70$), F4 (items 3, 7, 13; $\alpha_4 = 0.71$), and F5, which did not have dominant items. This structure accounted for 60.94% of the total variance.

Due to the absence of factor loading on the F5, we decided to conduct another EFA by forcing the factorization to four latent factors. The KMO verified the sampling adequacy for the analysis ($KMO$ = .87; individual $KMO$ values ≥0.59 and ≤0.92). LF1 consisted of seven items (items 1, 2, 4, 6, 9, 13, 14; $\alpha_1$ = 0.88), LF2 consisted of five items (items 5, 8, 10, 18, 19; $\alpha_2$ = 0.51), LF3 consisted of two items (items 3, 7; $\alpha_4$ = 0.64), and LF4 consisted of four items (items 11, 12, 15, 17; $\alpha_3$ = 0.70), which accounted for 55.27% of the total variance. Parameter estimates from the EFA are presented in Table 3. This reallocation was conceptually relevant, since LF1 (5 items of DIF and 2 items of DDF) referred to difficulty in awareness of feelings, LF2 (five items of EOT) referred to EOT, LF3 (two items of DIF) referred to difficulty in interoceptive capacities, and LF4 (3 items of DDF and 1 item of EOT) referred to poor affective sharing (Table 3).

In both studies, the four resultant factors of our analyses seemed to evaluate *a difficulty in awareness of feelings*, an EOT *interoceptive capacities, and a poor affective sharing*. The only main difference between Study 1 and Study 2 concerned items 4 and 11. Item 4, which refers to the capacity to describe one's own feelings, loaded on LF4 (*poor affective sharing*) in Study 1 and on LF2 EOT in Study 2. The opposite pattern was observed for item 11, which refers to the capacity to describe one's feelings about others. Moreover, both items 5 and 8 loaded on LF2 in Study 1, whereas they belonged to a separate factor with a low Cronbach alpha in Study 2. Therefore, it would have been statistically correct to suppress those items in Study 2, but we decided to keep them for theoretical reasons. Indeed, their meaning clearly reflects an external oriented thinking style, which is one of the features of alexithymia. However, the analyses showed that item 8 had a low correlation with the other items of LF2 in Study 1 (0.20) and in Study 2 (0.33). This item may therefore only be slightly representative of the alexithymia trait.

## Predictive value of the latent factors

The results of the four multivariate logistic regression analyses by study are reported in Table 4. When entering all LFs as predictors, the models with both somatic disorders (Study 1: $\chi^2(4)$ = 11.09, $p$ = 0.026, Nagelkerke $R^2$ = 0.14; Study 2: $\chi^2(4)$ = 22.38, $p$ < 0.001, Nagelkerke $R^2$ = 0.26) and medication intake (Study 1: $\chi^2(4)$ = 13.71, $p$ = 0.008, $R^2$ = 0.10; Study 2: $\chi^2(4)$ = 12, $p$ = 0.017, $R^2$ = 0.08) were significant in both studies. For eating disorders, the model was significant in Study 1 ($\chi^2(4)$ = 11.07, $p$ = 0.026, $R^2$ = 0.09) and approched significance in Study 2 ($\chi^2(4)$ = 9.17, $p$ = 0.057, $R^2$ = 0.08). For cardiovascular diseases, the model was only significant in Study 2 (Study 1: $\chi^2(4)$ = 3.40, $p$ = 0.493, $R^2$ = 0.04; Study 2: $\chi^2(4)$ = 10.98, $p$ = 0.027, $R^2$ = 0.12). LF1 positively predicted eating disorders in only Study 1. LF2 was not a predictor of any of the parameters. In both studies, LF3 was positively predictive of somatic disorders and medication intake, while in Study 2 it positively predicted eating disorders and cardiovascular diseases. Finally, LF4 negatively predicted medication intake in Study 1, although it was a trend in Study 2.

The results of the five multivariate linear regression analyses by study are reported in Table 5. When entering all LFs as predictors, the models that included the anxiety trait score (Study 1: $F(4,248)$ = 32.10, $p$ < 0.001, $R^2$ = 0.34; Study 2: $F(4,282)$ = 33, $p$ < 0.001, $R^2$ = 0.32), the depression score (Study 1: $F(4,248)$ = 23.16, $p$ < 0.001, $R^2$ = 0.27;

**Table 4 Detailed results of the multivariate logistic regression analyses.**

| | | Study 1 | | | | Study 2 | | | |
| --- | --- | --- | --- | --- | --- | --- | --- | --- | --- |
| | | Somatic disorders | Eating disorders | Medication intake | Cardiovascular diseases | Somatic disorders | Eating disorders | Medication intake | Cardiovascular diseases |
| Latent Factor 1 (LF1) | B | −0.10 | 0.13 | 0.01 | 0.07 | −0.07 | −0.06 | 0.04 | 0.06 |
| | Wald | 1.19 | 5.64 | 0.05 | 0.78 | 1.01 | 1.53 | 1.1 | 0.89 |
| | Exp(B) | 0.9 | 1.14 | 1.01 | 1.07 | 0.94 | 0.94 | 1.04 | 1.06 |
| | p | 0.276 | 0.018* | 0.820 | 0.376 | 0.315 | 0.216 | 0.294 | 0.346 |
| Latent Factor 2 (LF2) | B | −0.06 | −0.05 | −0.01 | −0.14 | −0.06 | −0.03 | −0.07 | 0.08 |
| | Wald | 0.23 | 0.32 | 0.01 | 1.46 | 0.28 | 0.121 | 1.05 | 0.56 |
| | Exp(B) | 0.94 | 0.95 | 0.99 | 0.87 | 0.94 | 0.97 | 0.93 | 1.08 |
| | p | 0.632 | 0.570 | 0.934 | 0.227 | 0.597 | 0.728 | 0.305 | 0.454 |
| Latent Factor 3 (LF3) | B | 0.62 | 0.14 | 0.39 | 0.17 | 0.74 | 0.33 | 0.22 | 0.34 |
| | Wald | 11.40 | 0.86 | 9.06 | 0.75 | 15.68 | 8.72 | 5.40 | 5.22 |
| | Exp(B) | 1.86 | 1.15 | 1.47 | 1.18 | 2.09 | 1.40 | 1.25 | 1.41 |
| | p | 0.001*** | 0.354 | 0.003** | 0.387 | <0.001*** | 0.003** | 0.020* | 0.022* |
| Latent Factor 4 (LF4) | B | 0.01 | −0.03 | −0.17 | −0.001 | −0.13 | 0.01 | −0.12 | −0.10 |
| | Wald | 0.004 | 0.08 | 3.87 | 0.00 | 1.45 | 0.01 | 3.38 | 1.05 |
| | Exp(B) | 1.01 | 0.97 | 0.84 | 1 | 0.88 | 1.01 | 0.89 | 0.9 |
| | p | 0.951 | 0.781 | 0.049* | 0.995 | 0.228 | 0.941 | 0.066[t] | 0.306 |

Notes:
[t] $p < 0.07$.
* $p < 0.05$.
** $p < 0.01$.
*** $p < 0.001$.

**Table 5 Detailed results of the multivariate regression analyses.**

| | | Study 1 | | | | | Study 2 | | | | |
| --- | --- | --- | --- | --- | --- | --- | --- | --- | --- | --- | --- |
| | | STAI-T | HADS-D | BFI-N | PSS | F–D | STAI-T | BDI-13 | BFI-N | PSS | F–D |
| Latent Factor 1 (LF1) | B | 1.17 | 0.22 | 0.72 | 0.73 | −0.09 | 0.67 | 0.23 | 0.42 | 0.45 | −0.10 |
| | t | 8.13 | 4.74 | 6.40 | 6.17 | −3.36 | 7.04 | 4.88 | 5.87 | 5.81 | −6.00 |
| | p | <0.001*** | <0.001*** | <0.001*** | <0.001*** | 0.001*** | <0.001*** | <0.001*** | <0.001*** | <0.001*** | <0.001*** |
| Latent Factor 2 (LF2) | B | 0.32 | 0.21 | 0.07 | 0.19 | −0.23 | −0.14 | −0.06 | −0.04 | −0.03 | −0.06 |
| | t | 1.56 | 3.23 | 0.46 | 1.13 | −5.71 | −0.79 | −0.64 | −0.32 | −0.21 | −1.95 |
| | p | 0.120 | 0.001*** | 0.646 | 0.261 | <0.001*** | 0.433 | 0.523 | 0.750 | 0.833 | 0.053[t] |
| Latent Factor 3 (LF3) | B | 0.67 | 0.31 | 0.56 | 0.65 | −0.15 | 0.98 | 0.40 | 0.67 | 0.57 | −0.10 |
| | t | 1.72 | 2.53 | 1.83 | 2.02 | −1.93 | 3.96 | 3.21 | 3.66 | 2.84 | −2.42 |
| | p | 0.088 | 0.012* | 0.068[t] | 0.045* | 0.054[t] | <0.001*** | 0.001*** | <0.001*** | 0.005** | 0.016* |
| Latent Factor 4 (LF4) | B | −0.20 | 0.03 | −0.36 | −0.03 | 0.01 | −0.16 | 0.06 | −0.26 | 0.02 | 0.03 |
| | t | −0.94 | 0.50 | −2.15 | −0.20 | 0.17 | −1.06 | 0.75 | −2.34 | 0.19 | 1.30 |
| | p | 0.346 | 0.618 | 0.033* | 0.844 | 0.868 | 0.290 | 0.451 | 0.020* | 0.846 | 0.195 |

Notes:
[t] $p < 0.07$.
* $p < 0.05$.
** $p < 0.01$.
*** $p < 0.001$.
STAI-T, State-Trait Anxiety Inventory; HADS-D, Depression subscale of the Hospital Anxiety and Depression Scale; BDI-13, Beck Depression Inventory-13; BFI-N, Neuroticism dimension of the Big Five Inventory; PSS, Perceived Stress Scale; F–D, Difference score between functional and dysfunctional coping from Brief Cope.

Study 2: $F(4,282) = 21.03$, $p < 0.001$, $R^2 = 0.23$), the emotional instability score (Study 1: $F(4,248) = 17.25$, $p < 0.001$, $R^2 = 0.22$; Study 2: $F(4,282) = 22.12$, $p < 0.001$, $R^2 = 0.24$), the perceived stress score (Study 1: $F(4,248) = 21.91$, $p < 0.001$, $R^2 = 0.26$; Study 2: $F(4,282) = 23.67$, $p < 0.001$, $R^2 = 0.25$), and the coping difference score (Study 1: $F(4,248) = 21.47$, $p < 0.001$, $R^2 = 0.26$; Study 2: $F(4,282) = 21.97$, $p < 0.001$, $R^2 = 0.24$) were significant in both studies. In both studies, LF1 was positively associated with all measures (anxiety, depression, emotional instability, perceived stress, and effective coping strategies). LF2 was positively associated with depression scores and negatively associated with effective coping strategies in Study 1, whereas LF2 approached significance in the negative prediction of effective coping strategies in Study 2. LF3 was positively associated with depression and perceived stress scores in both studies. LF3, however, was also positively associated with anxiety and emotional instability scores, and negatively associated with effective coping strategies in Study 2, and approached significance in the positive prediction of the emotional instability scores and in the negative prediction of effective coping strategies in Study 1. Finally, LF4 was negatively associated with emotional instability scores in both studies. We discuss those relationships in the general discussion.

The results of the regression analyses are reported in Fig. 1.

## General discussion

The aim of the two studies was to examine the existence of a potential latent factor in the TAS-20 structure. We also examined its external validity by investigating the relationships of the new latent alexithymia factors with psychological, physical health and personality trait measures. As expected, our results mainly highlight the presence of a new latent factor in the assessment of alexithymia, which seems to reflect interoceptive abilities.

The EFA performed on the TAS-20 showed the existence of factors that did not strictly refer to the theoretical factors mentioned in the literature (*Bagby, Parker & Taylor, 1994*; *Taylor, Bagby & Parker, 2003*). Among these factors, items 16 and 20 had very low internal reliability. These items represent the preference for entertainment rather than an exploration of a deeper meaning in movies or plays, which appears to reflect social norms rather than a core alexithymia trait (*Dere et al., 2013*). Moreover, previous studies have found that items 16 and 20 correlate weakly or not at all with the EOT subscale, suggesting that these items would not be the ideal candidates for the assessment of alexithymia (*González-Arias et al., 2018*; *Kooiman, Spinhoven & Trijsburg, 2002*). Therefore, we decided to remove them from the scale, which resulted in a new TAS consisting of 18 items. Furthermore, while items 5, 8, 10, 18, and 19 loaded on the same factor in Study 1, both items 5 and 8 emerged as separate single factors in Study 2, with very low internal reliability. Indeed, items 5 and 8 focus on problem-solving whereas the remaining items (i.e., items 10, 18, 19) focus on emotions, which could explain why they did not load on the same factor in the first EFA in Study 2. These two items, however, represent a concrete cognitive style and seem to be representative of the alexithymia trait. Based on these considerations, suppressing them would have been against the theory, so we decided to retain them. Despite this, correlations between item 8 and the other items loading on the

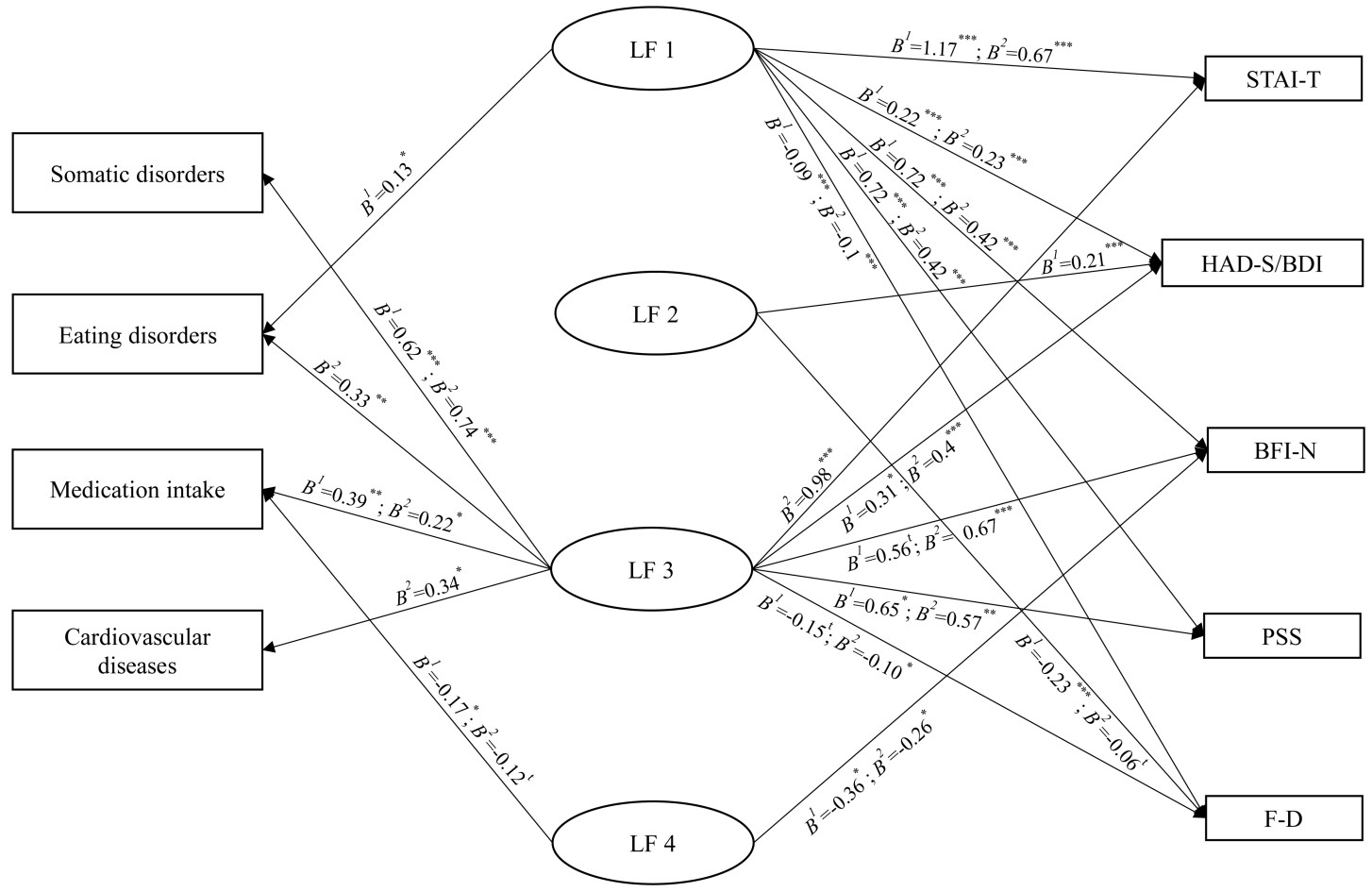

**Figure 1 Associations between the latent factors and psychological and physical health measures.** *Notes.* [t]$p < 0.07$, [*]$p < 0.05$, [**]$p < 0.01$, [***]$p < 0.001$. $B^1$ are the results of the analyses of Study 1 and $B^2$ are the results of the analyses of Study 2.

same factor were still low (<0.30), so it may be more appropriate to reconsider its place in the evaluation of alexithymia or to rewrite it. In addition, because the aim of this paper was not to validate or confirm the factor structure of this scale but to explore whether the TAS-20 contains a latent factor assessing interoceptive abilities, we will not discuss the validity of this factor. However, it is possible that the presence of items 16 and 20 may explain the lack of internal consistency of this dimension (*Bressi et al., 1996*; *Taylor, Bagby & Parker, 2003*; *Zhu et al., 2007*).

Our results mainly highlight the presence of new latent factors in the assessment of alexithymia. Interestingly, those factors were present in an earlier structure of the TAS, originally called TAS-26 (*Taylor, Ryan & Bagby, 1985*). We found that in the present structure, LF1 (*difficulty in awareness of feelings*) always included items from the DIF/DDF dimensions of the TAS-20, and all of these items belonged to the previous *capacity to identify and to distinguish between feelings and bodily sensations* present in the TAS-26. Furthermore, LF2 included the same items from the EOT dimension in both studies, three of which previously belonged to the old *preference for focusing on external events rather than inner experiences* dimension of the TAS-26. The remaining two items were created

after the TAS-26 review. This clustering is consistent with previous works supporting an oblique two-dimensional model in which DIF and DDF belonged to the same factor while EOT formed a single factor (*Erni, Lötscher & Modestin, 1997*; *Kooiman, Spinhoven & Trijsburg, 2002*; *Loas et al., 1996*). Moreover, the authors also proposed this clustering for the TAS-R version (*Taylor, Bagby & Parker, 1992*). We found associations between LF1 (*difficulty in awareness of feelings*, grouping DIF and DDF items) and all psychological outcomes, emotional instability and with the use of copying strategies that are mostly maladaptive. High scores in the DIF or DDF dimensions can be associated with poor emotion regulation, which is linked with mental health alterations (*Luminet & Zamariola, 2018*). *Cutuli (2014)* proposed that higher scores in the DDF dimension are associated with higher levels of expressive suppression, which is considered as a maladaptive regulation strategy, and that this strategy is associated with stronger depressive symptoms, lower interpersonal functioning, and decreased levels of well-being. In addition, higher scores in the DIF factor were related to limited use of emotion regulation strategies and the DDF or DIF-depression relationships were mediated by experiential avoidance, a tendency to avoid negative internal experiences (sensations, emotions, thoughts, memories) (*Hayes et al., 2004*). In the present paper, these associations between LF1 and psychological outcomes support the idea of an ineffective emotion regulation in individuals suffering from an overall decrease in awareness of feelings. Moreover, the negative association of LF2 with effective coping strategies in both studies suggests that it is difficult for alexithymic people to cope with difficult situations, which reflects their deficits in emotion regulation (*Luminet & Zamariola, 2018*). These results are highly consistent with the literature, since deficits in emotion regulation have been well documented in people with alexithymia (*Luminet & Zamariola, 2018*; *Lumley, Neely & Burger, 2007*). In spite of this empirical evidence, grouping DIF and DDF dimensions was not consistent in the literature, since this clustering depends on the types of statistical tools chosen (EFA vs. CFA), although a three-dimensional model (i.e., DIF, DDF, EOT) is still considered as the best fit (*Loas et al., 2001*).

As expected, our results confirmed the existence of a new latent factor in the assessment of alexithymia, which seems to reflect interoceptive abilities. The LF3 (*difficulty in interoceptive abilities*) included two items (items 3[1] and 7[2]) from the DIF dimension of the TAS-20. These are the only items that explicitly refer to physical and bodily sensations and therefore reflect the presumed clustering of awareness of feelings and interoceptive abilities. In the TAS-26, these items were again part of the *capacity to identify and to distinguish between feelings and bodily sensations*. Even if later scale development of the TAS excluded the specific assessment of difficulty in distinguishing between feelings and bodily sensations, some items, which still evaluated this feature (items 3 and 7), were included in the TAS-20 review. The present research thus supports the existence of an independent latent dimension permitting the assessment of this ability. This finding is quite consistent with the literature indicating an *atypical* interoception in alexithymic individuals (*Murphy et al., 2017*). Interoception is composed of three dimensions:
(i) interoceptive sensibility (IS) (subjective abilities to report on body states);
(ii) interoceptive accuracy (IAcc) (objective abilities to perceive internal body changes);

[1] Item 3: *"I have physical sensations that even doctors don't understand".*

[2] Item 7: *"I am often puzzled by sensations in my body".*

(iii) interoceptive awareness (degree of overlap between IAcc and IS) (*Pollatos & Herbert, 2018*). A fourth dimension has recently been proposed, the emotional evaluation of interoceptive signals. This dimension refers to the emotional degree attributed to the bodily sensations that are expressed or taken into account in a specific situation (*Pollatos & Herbert, 2018*). The study of the relationship between alexithymia and dimensions of interoception is currently receiving considerable attention. However, the studies are contradictory. Some studies show a link between alexithymia and IAcc (*Herbert, Herbert & Pollatos, 2011*; *Murphy, Catmur & Bird, 2018*; *Shah et al., 2016*) and IS (*Brewer, Cook & Bird, 2016*), while others show a weak correlation between IS and alexithymia (*Zamariola et al., 2018*) and no relationship with IAcc (*Bornemann & Singer, 2017*; *Zamariola et al., 2018*). One explanation for this disparity could be the choice of tools. Currently, the validity of the task most commonly used to measure IAcc, the Heartbeat Tracking task (*Schandry, 1981*), is being questioned by various authors (*Desmedt, Luminet & Corneille, 2018*; *Ring et al., 2015*). In addition, this task focuses on only the perception of heart rate and does not consider the ability to perceive other internal body sensations of interoception. Also, IS measures face the usual limitations of self-reporting and the questionnaires used to measure these abilities are very different from one study to another (*Brewer, Cook & Bird, 2016*; *Zamariola et al., 2018*). Moreover, in view of the present results, we can speculate that alexithymic individuals do not present difficulties in perceiving or reporting internal body sensations, but rather a difficulty in interpreting body sensations. This could correspond to the fourth dimension proposed by *Pollatos & Herbert (2018)*, the emotional evaluation of interoceptive signals, and would be consistent with the positive relationship between alexithymia and a perception of similarity between emotional and non-emotional states (*Brewer, Cook & Bird, 2016*). New studies are needed to further develop this hypothesis. As observed in alexithymia, such an atypical functioning, in association with an alteration of emotional awareness, could lead in the long run to the development of psychosomatic diseases (*Kanbara & Fukunaga, 2016*; *Porcelli & Taylor, 2018*). It also corroborates our results that showed a positive association between this dimension and the presence of somatic diseases. In both studies, individuals with high scores on LF3 (*difficulty in interoceptive awareness*) were more likely to exhibit somatic disorders and to take medications compared to those with low scores. Study 2 showed that LF3 could also be related to cardiovascular diseases. To our knowledge, few studies have shown the link between coronary heart disease and interoception (*Kollenbaum, 1994*). According to the present results, this study mentions that patients with coronary heart disease generally underestimate their heart rate. However, high levels of alexithymia are found in patients with cardiovascular disease, particularly those with hypertension (*Porcelli & Taylor, 2018*). Since the TAS-20 contains a latent factor for measuring interoceptive abilities, it may be that these associations with alexithymia are due to the presence of this latent factor. Further studies are needed to further develop this hypothesis. Regarding psychological issues, individuals with high scores for LF3 were more likely to exhibit eating disorders, high emotional instability scores, and dysfunctional coping strategies. Therefore, it is important not to neglect the evaluation of this interoceptive dimension, considering that it could allow the referral of alexithymic

individuals to the appropriate therapies in the fields of somatic and psychological health. In addition to promoting the recognition and regulation of feelings for individuals with high scores in LF1 (*Thoma & Greenberg, 2015*), proposing therapies based mainly on the processing of interoceptive signals emanating from the body could constitute a new perspective for preventive health programs in patients with high LF3 scores. With this in mind, the LF3 subscale would benefit from supplementary items dealing with interoception.

LF4 (*poor affective sharing*) included items from the DDF/EOT dimensions of the TAS-20. Two of them belonged to the DDF dimension of the TAS-26, which refers to the ability to communicate feelings to other people, and the other items were created during the TAS-26 review. The main difference between Study 1 and Study 2 concerned items 4 and 11. In Study 1, item 4 belonged to LF4 and item 11 to LF1 while in Study 2, it was the opposite. Interestingly, these two items also belonged to two axes in TAS-26. They were used to assess both the *capacity to identify and to distinguish between feelings and bodily sensations* and the *ability to communicate feelings to other people*. These two items may, therefore, be ambiguous, even if they belonged to the DDS dimension in TAS-20. Finally, LF4 (*poor affective sharing*) was negatively associated with emotional instability scores and medication intake. Since we performed multivariate regression analyses, the predictive effect of LF4 was analyzed in the unique context of social-affective sharing, thereby controlling for the effect of the other LFs. The predictive effect of LF4 might, therefore, reveal that the least emotionally stable individuals are less likely to feel the need to share their emotions and affects with others. In such a context, high scores in LF4 could predict low levels of emotional instability and medication intake.

## Limits

Some limitations should be mentioned. First, the four health measures were collected using binomial self-report questions. It would be interesting to replicate our results using structured interviews and/or validated questionnaires. Second, the recruited population was a subclinical population, so it was not possible to generalize about the relationship between alexithymia and clinical symptomatology. This population was composed of individuals from the general population and students. While in Study 1 we had both individuals from the general population and students, in Study 2 we had only students. Additionally, we used two different depression scales and two different alexithymia scales, although there are no significant differences between the two versions apart from the Likert scale. As a consequence, the groups are difficult to compare. Third, we had to recruit students to complete our sample for Study 1 because we could not get enough participants from the general population. Finally, in this paper, the studies conducted are transversal and not longitudinal. Thus, we cannot conclude if alexithymia is a common co-occurring or a primary pathology. The impact of alexithymia on health and its etiology are difficult to understand because it requires longitudinal studies to know its causal relationships and origin. Some authors postulate that alexithymia is an inherited trait, others mention that alexithymia follows a deficit in the development of affects during childhood, or even postulate that it is a defense mechanism put in place to deal with

negative emotions that are difficult for individuals to overcome. However, a multifactorial etiology is strongly suggested and could be the result of a combination of environmental and genetic factors (for review see *Taylor & Bagby, 2013*). From a clinical perspective, it is important to know the etiology of alexithymia and its role on health. To ensure this, longitudinal studies must be considered.

## CONCLUSIONS

Despite the fact that we used TAS-20 versions measured using a 4-point scale in Study 1 and on a 5-point scale in Study 2, we found a very similar distribution of items across both studies. The latent structure of the TAS-20 reflects a substantial part of the older structure of the scale. Strikingly, one of those latent factors is linked to an important concept from the TAS-26: the interoceptive abilities. Its associations with somatic issues highlight the key role of the body awareness component in alexithymia, which is currently neglected in the evaluation of this construct. The alexithymia scale with a full dimension covering interoceptive abilities would open new possibilities in the research field of alexithymia. From a health perspective, this could also contribute to better management of alexithymic individuals, as it would allow health professionals to refer them to the most appropriate preventive therapies.

## ACKNOWLEDGEMENTS

The authors are grateful to the participants who took part in the studies.

### Funding

This work was supported by the Auvergne Region (MONDILLON 2013, 13-CER-139), France. The funders had no role in study design, data collection and analysis, decision to publish, or preparation of the manuscript.

### Grant Disclosures

The following grant information was disclosed by the authors:
Auvergne Region: MONDILLON 2013, 13-CER-139, France.

### Competing Interests

The authors declare that they have no competing interests.

### Author Contributions

- Alicia Fournier conceived and designed the experiments, performed the experiments, analyzed the data, prepared figures and/or tables, authored or reviewed drafts of the paper, approved the final draft.
- Olivier Luminet authored or reviewed drafts of the paper, approved the final draft.
- Michael Dambrun authored or reviewed drafts of the paper, approved the final draft.
- Frédéric Dutheil authored or reviewed drafts of the paper, approved the final draft.
- Sonia Pellissier authored or reviewed drafts of the paper, approved the final draft.

- Laurie Mondillon conceived and designed the experiments, authored or reviewed drafts of the paper, approved the final draft.

## Human Ethics

The following information was supplied relating to ethical approvals (i.e., approving body and any reference numbers):

The Ethics Committee in Clermont-Ferrand approved the study protocol (CPP SUD-EST 6, IRB00008526, 2015-CE23).

## Data Availability

The experimental data is available at OSF: https://osf.io/8kncz.

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
