# Peer review of "Importance of considering interoceptive abilities in alexithymia assessment"

_PeerJ, doi:10.7717/peerj.7615_

## Round 0.1 · original submission · Major Revisions

Three reviewers have now read your manuscript and they each have suggested revisions. Please address these issues and resubmit.

Reviewer 1 ·

Basic reporting

1) Abstract, line 27: Do you mean that early versions of the TAS contained items pertaining to interoception that are no longer included? This sentence is confusing and later sentences seem to contradict what I concluded from it.

2) Abstract, lines 29 – 30: Grammatical errors: “However, the revised version of alexithymia scale, the TAS-2-, contains three factors solution that do not involve interoceptive abilities.”

3) Abstract, line 33: I would note that these only include self-reported measures of psychological and physical health.

4) Abstract, line 44: It should be noted that there was a positive associated only with all measured psychological issues. The testing battery was not exhaustive of all possible psychological issues. The phrase ‘psychological issues’ is also rather ambiguous. Does this refer to clinical disorders, self-reported concerns or something else?

5) Abstract, line 48: What does “it seems important to develop a conceptual justification for the assessment of interoceptive abilities when considering the evaluation of alexithymia” mean?

6) Introduction, line 53: Page numbers should be provided with quotes.
Introduction, line 70: Could the authors identify the scales that were previously identified as lacking adequate psychometric properties?

7) Introduction, line 85: The phrase “fully-fledged dimension” should be replaced with more scientific terminology.

8) Introduction, line 85: The authors have thus far discussed factors and items within the structure of the TAS. What does a label constitute?

9) Introduction, line 89: Do the authors mean that the TAS-20 is the most widely used measure of alexithymia in empirical research, not the most widely used scale, generally-speaking?

10) Introduction, line 100: the TAS-20 is not designed for use in children. There are other measures based on the TAS-20 that have been validated in children. See, for example, the Alexithymia Questionnaire for Children (Rieffe et al., 2006, Personality and Individual Differences).

11) Introduction, line 128: “psychological and physical troubles” is a rather broad, colloquial term. Greater precision of terminology may be warranted.

12) Methods, line 141: “The remaining of participants” is grammatically incorrect.

13) Results, line 231: The authors report that items 16 and 20 were omitted because they loaded onto a factor with low reliability. It would be helpful to state what these items were here.

14) Discussion, line 392: “difficulty in awareness feelings” is not grammatically correct.
Discussion, line 404: Please rephrase “a massive deficit” with more appropriate language.

15) A figure showing the associations between the alexithymia factors and psychological and physical health measures would be helpful.

Experimental design

1) In this study, the authors investigate the presence of a latent ‘interoceptive ability’ factor in the Toronto Alexithymia Scale, and how this relates to indicators of psychological and physical health, broadly defined. To this end, the authors employ a general population sample (predominately late adolescent, majority female psychology undergraduates) and questionnaire measures of various psychological and physical health characteristics, as well as measures of personality traits. Generalizations are then made about clinical symptomatology. There seems to be some confusion around the interpretation of personality traits as indicative of psychological and physical disorder, see for example, inclusion of the Neuroticism dimension of the Big Five personality inventory. Greater explanation of how these measures relate to clinical symptomatology should be provided, including reporting how many of the participants in this study meet cut-off criteria for clinically significant symptoms. Alternatively, the manuscript could be revised to more clearly reflect that this is study of subclinical traits in the general population, with adequate description of the limitations of generalizing from such a design.

2) Introduction, line 134: “The results of both studies support the importance of considering the presence of this latent factor”. Which two studies? In the absence of references, I assume the authors mean the two studies in the present manuscript here. It is rather circular to justify the present study with the results of that study.

3) In general, hypotheses should be outlined more clearly.

4) Have any important differences been identified been the two versions of the TAS-20 (4-point Likert scale and 5-point Likert scale)? Could the authors speak more to the justification for including both in the present design?

5) Introduction, line 130: Could the authors expand on why they propose that factors of the TAS-20, apart from the proposed latent interoception factor, will not also relate to physical and psychological healthy problems? Are you proposing that previously observed associations between alexithymia and ill health are fully mediated by interoception, or that interoception and alexithymia are one and the same?

6) Methods, line 142: Based on the numbers, it looks as though Study 2’s sample was comprised only of undergraduate students, and Study 1’s sample was comprised of both undergraduates and general population adults recruited through social networks. Please explicitly state how these samples were formed and why the recruitment strategies differed.

7) Methods, line 168: The Hospital Anxiety and Depression Scale (HADS-D; Zigmond & Snaith, 1983) is a slightly odd choice for measuring depression symptoms in a general population sample. This tool was originally designed for use in medical outpatient clinics and not the general population. What was the motivation for using this scale in Study 1, and what motivated the switch to the Beck Depression Inventory in Study 2?

8) Methods, line 199: has the difference score between functional and dysfunctional coping been previously validated, or is it novel to this study? How does it distinguish between high levels of functional coping and low levels of dysfunctional coping versus high levels of dysfunctional coping and low levels of functional coping?

9) Methods, Line 202: In the absence of using validated measures of somatic disorders, eating disorders etc please provide further details. For example, were these questions designed to uncover if participants had a prior history of these conditions, or if they exhibited specific behaviors associated with these conditions? Consider including the questions in the supplementary materials so that readers can evaluate and/or fully replicate the design. How were scores on the questions treated during analysis, summed, averaged etc? Scores appear to be binary due to the use of logistic regression, but this should be stated explicitly.

10) Methods, line 205: How was the information on medication usage used? Were different classes of medication collapsed in the multiple regression analysis? Please report what medications were reported and their frequency.

Validity of the findings

1) Results, line 332: The description of the multiple regression analyses is very confusing. Here’s it is stated that ‘NFs’ are entering as predictor variables but in Table 4 it appears that the TAS-20 factors are entered as the dependent variable (one model per factor) and the health variables are entered as the predictors. However, in the text, it reads as though one regression model has been conducted for each health variables as well (e.g. “For cardiovascular diseases, the model was significant..” with TAS-2- factors as predictor variables. Please clarify exactly how many regression analyses were performed. Further, is there an acceptable lack of multicollinearity to enter all the TAS-20 factors as predictor variables in the regression models? Relatedly, what does ‘NF’ stand for? In Table 4, the acronym ‘LF’ appears to be used, indicating latent factor.

2) Results, line 246: how was the “trend for significance” alpha level set?
Results, line 296: Please expand on the statement “omitting them would be against the theory”.

3) L. 388: Please expand on what is meant by the statement “it may be more appropriate to reconsider its place in the evaluation of alexithymia or to rewrite it”. It is not clear what is meant here.

4) Discussion, line 407: Could the authors expand on the link with emotion regulation and how this relates to the present study? Moreover, it is not clear how the association with emotion regulation relates to the grouping of DUF and DDF factors.

5) Discussion, line 432: The association between the new interoceptive dimension and cardiovascular disease warrants further discussion. Are the authors aware if this has previously been reported in the literature? Can they offer a rationale for why this relationship might exist?

6) Results, line 325: Here the authors state that an external oriented thinking style is one of the key features of alexithymia. However, in the Introduction they appeared to question the validity of this factor. Could they clarify.

7) Much of the discussion takes the stance that individuals high in alexithymia are at risk for developing physical and psychological symptoms of ill health. However, could it also not be possible that alexithymia represents a common co-occurring feature in many health conditions, and not necessarily the primary pathology? Likewise, could it be that individuals with ill health then develop alexithymia?

Additional comments

1) Abstract, line 26: The authors state that “high alexithymia scorers have deficits in their interoceptive abilities which can lead to psychological and physical disorders”. This language implies a causality between interoception and illness that is yet to be firmly established. Further, what constitutes a deficit in interoception? Measures of interoception show variability across the population, with the majority of healthy adults being quite poor at objectively detecting their interoceptive signals. Is reduced interoception necessarily an indicator of impaired function?

Reviewer 2 ·

Basic reporting

The aim of the paper is to re-examine the factorial structure of the TAS-20 using exploratory factor analyses (EFA) and then investigate the relationships between the resulting factors and other psychological and health data using regressions. To this end, authors conducted two online studies (N=253 and N=287). Results show a new structure of the questionnaire comprising four, rather than three, factors. Specifically, this structure reveals a new latent factor that authors name difficulty in interoceptive abilities. Authors conclude that this factor should be included in future assessment of alexithymia.

Overall, the paper is clearly written and the topic under investigation appears timely and relevant for the field of alexithymia. I have a few points that I believe authors should address to improve the current manuscript before publication.

The interoceptive difficulties that characterize individuals with alexithymia are an aspect that is currently receiving significant attention in the field, as shown by several papers published in the last years (e.g. J Murphy, R Brewer, H Hobson, C Catmur, G Bird - Biological psychology, 2018; J Murphy, C Catmur, G Bird Journal of Experimental Psychology: General, 2018; R Brewer, R Cook, G Bird Royal Society Open Science, 2016; C Scarpazza, H Huang, A Zangrossi, S Massaro, Journal of Psychosomatic Research 2018; C Scarpazza, G di Pellegrino Current Developments in Alexithymia-a Cognitive and Affective Deficit, 2018; Herbert BM, Herbert C, Pollatos O., Journal of personality. 2011 Oct;79(5):1149-75.). Nevertheless, this literature is not mentioned in the paper. Authors should include these studies in their introduction in order to strengthen the rationale for their study. Additionally, they should enrich their discussion by commenting their results in the broader context of this literature.

Experimental design

Methods:
Regarding the coping strategies scores, I believe authors should clarify the rationale used to classify each coping strategy in each of the three categories, especially for those placed in the “coping with varying functionality”. Additionally, it seems that the mean score for the “coping with varying functionality” category was neither used to compute the coping strategy score nor as dependent variable in the regression model. Was this an a priori choice? If so, why was data collected also on those scales and why were they not included in the analyses? Please, clarify these points.

Validity of the findings

Results:
I suggest authors to report the direction of significant regressions for all the significant regressions, because it would ease readability. This is done for most but not for all regressions (e.g. line 338 “LF1 predicted eating disorders only in Study 1”).

Line 355-356 LF3 is stated to positively predict effective coping strategies in Study 1. I believe it should be “negatively”.

Additional comments

Minor issues:
Line 106 should state “other solutions” instead of “others solutions”

·

Basic reporting

no comment

Experimental design

no comment

Validity of the findings

no comment

Additional comments

My thanks to Fournier and colleagues for their work on this project. As I understand it, you have argued that the TAS-20 used to assess trait varience in emotional insight contains a factor which was purportedly removed when the TAS-26 was updated. You also assessed the validity of the 4 factors of TAS-20 by examining their power in predicting several health outcomes. I liked this study and I think that this kind of work can be underappreciated. I recommend that it be published.

I recommend just two small additions:
1. In the introduction section could you add a sentence or two on the theoretical or practical significance of imposing an incorrect factor structure on the TAS-20 (if indeed the 3 factor solution is misleading).
2. In the procedure could you indicate where in the sequence of surveys subjects answered the demographic items.

---

## Round 0.2 · accepted · Accept

Please address the final minor suggestions while in production.

Reviewer 1 ·

Basic reporting

I thank the authors for the significant revisions they have made to their manuscript, which is now much stronger. I have one remaining suggestion, but otherwise would recommend this manuscript for publication.

Line 589: In the discussion of the limitations of the study, the word “transversal” should be replaced with “cross-sectional”, as this is the correct terminology for what I believe the authors are trying to say.

Experimental design

No comment

Validity of the findings

No comment

Reviewer 2 ·

Basic reporting

I believe authors exhaustively addressed my comments.

Experimental design

I believe authors exhaustively addressed my comments.

Validity of the findings

I believe authors exhaustively addressed my comments.

·

Basic reporting

meets standards

Experimental design

meets standards

Validity of the findings

meets standards

Additional comments

My thanks to Fournier and colelagues for their revisions. I am now happy to recommend acceptance.
best,
Glenn Carruthers